# Differential Impact Analysis for Climate Change Adaptation: A Case Study from Nepal

**Chiranjeewee Khadka** [1,*], **Anju Upadhyaya** [2], **Magda Edwards-Jonášová** [1], **Nabin Dhungana** [3], **Sony Baral** [2] **and Pavel Cudlin** [1]

1 Department of Carbon Storage in the Landscape, Czech Academy of Sciences, 11000 Prague, Czech Republic
2 Institute of Forestry, Tribhuvan University, Kathmandu 44600, Nepal
3 Department of Natural Resources and Environmental Studies, College of Environmental Studies, National Dong Hwa University, Hualien 97401, Taiwan
* Correspondence: khadka.c@czechglobe.cz

**Abstract:** Following a case study, community adaptation plans are a bottom-up approach that focus on increasing climate-vulnerable communities' engagement in local adaptation planning and policy design, prioritization, and implementation in Nepal. This paper explains how Community-Based Adaptation Action Plan (CAPA) groups are being studied to assess the climate vulnerability of the local socio-ecosystem and to develop community-level adaptation measures. However, there is insufficient research to differentiate local vulnerabilities caused by climate change. This paper, therefore, examines climate change vulnerability with respect to community vulnerability and potential adaptation measures to increase community resilience and adaptive capacity through CAPAs. The study compares differences by gender, caste/ethnicity, and wealth in relation to specific climate-related hazards (exposure, sensitivity, and adaptive capacity) of communities. The study draws on secondary sources of information along with field observations, 73 household interviews, 13 key-informant interviews, consultations, and 9 interactive meetings in 3 districts of Nepal. Differential impact analysis refers to the exposure, sensitivity, and adaptive capacity of local socio-ecological systems. In addition, multivariate analysis was conducted using the Canoco program to analyze the role of actors with respect to climate vulnerability. The results conclude that the degree of vulnerability varies widely at the household level and is strongly influenced by socio-economic characteristics such as gender, caste/ethnicity, and wealth. Immediate and focused attention is needed to improve access to government resources for vulnerable households, requiring positive support from decision makers. Equally important is improving the chain of communication, which includes information, skills, knowledge, capacity, and institutional arrangements. Analysis of the differential vulnerability and the adaptive capacity of a vulnerable community is more appropriate for the design of local adaptation plans. Therefore, the study suggests that engagement of local partners, including local authorities, in addressing vulnerability and adaptation is required to confront the social process, new institutional arrangements, local adaptation, and capacity-building with technical solutions.

**Keywords:** climate vulnerability; community-based adaptation; multi-variate analysis; gender; differential impact; Nepal

## 1. Introduction

In recent years, climate change adaptation plans have emerged as popular development agendas to address the vulnerabilities and negative impacts of climate change on human and natural systems. Adaptation plans are typically implemented for current and short-term timescales of vulnerability assessment and are more localized, such as to households or communities. Adaptation plans represent a way in which local individuals, households, and communities can change their mix of productive activities and modify their community rules and institutions in response to vulnerability to sustain their livelihoods [1]. Localized experience and observation-based knowledge is crucial with respect

to climate change adaptation science [2]. The adaptation process is necessary to understand vulnerable systems, drivers of vulnerability, differential vulnerability, and local adaptive capacities to address risks and resilience to the impacts of climate change [3,4]. Adaptation approaches are adapted to identify critical information regarding socio-economic vulnerabilities and opportunities, resource degradation, food scarcity, and provision of basic services related to climate change in each of the local sites [5]. Assessing vulnerability, exposure, sensitivity, barriers to adaptation actions, and adaptive capacity are necessary to identify and implement subsequent actions. Better understanding of the adaptive capacities of communities and the limits of adaptation is needed to strategize alternatives for adaptation [6–8]. Measuring vulnerability is considered a prerequisite for climate change adaptation [9]. Vulnerability is not an immutable state, but a multidimensional process influenced by social, political, and economic forces that interact from local to international scales [9,10]. Vulnerability is expressed as a function of exposure, sensitivity, and adaptive capacity scales [11]. It is a function the characteristics, magnitude, and speed of the climate change to which a system is exposed, its sensitivity, and the adaptive capacity of the system [6,12].

In the context of climate change, vulnerability and the existing differential impacts of vulnerability serve as a starting point for understanding vulnerability contexts and their exposure, sensitivity, and adaptive capacity to climate change [4,12–14]. Differential vulnerability is related to current climatic and geographic heterogeneity, as well as the diversity of social factors that influence vulnerability [15]. Vulnerability to climate change varies among different groups of people based on their position in social and gender structures in a particular location and at a particular time [16]. Vulnerability assessments compare vulnerability under different socio-economic conditions, climatic and non-climatic factors, and adaptive responses that relate the present to the future [13]. However, it is a difficult task to assess vulnerability; the different impacts of climate change at household and community levels are important to identify the most vulnerable groups to develop adaptation plans and strategies in terms of resource allocation in specific contexts and regions [5]. Climate change and its impact on natural and socio-economic factors in Nepal, particularly the local adaptation process, focuses mostly on coping with various types of hazards caused by extreme weather events due to climate change [17]. Therefore, the study refers to a social ecosystem state approach to assess the vulnerability contexts in the study regions based on exposure, sensitivity, and adaptive capacity to climate change.

The differential vulnerability of people to environmental hazards results from a range of social, economic, historical, and political factors, all operating at multiple scales [18]. The impacts of climate change (CC) are not gender-neutral, as they are not evenly distributed between women and men. The disparity varies according to poverty, access to resources, other economic wealth, and power relations. Although vulnerability is experienced locally, the causes of vulnerability are at different social, geographic, and temporal scales [19], with the poor, women, *Dalit*, *Janajati*, and excluded groups being more vulnerable [20]. Women suffer more than men from extreme events related to climate change such as droughts, floods, extreme rainfall, etc., due to their physical condition, systemic gender discrimination, and societal expectations related to gender roles [21]. Households that depend on natural resources such as water, forest products, etc., for their livelihoods have been identified as the most vulnerable communities in Nepal [22,23]. It is also clear that climate change does not affect women and men equally [24], with vulnerability to climate change impacts being higher among women within society and households due to gender differences [25]. In line with the IPCC and UNFCCC gender and climate change report, some authors argue that women tend to have fewer assets and rights than men, are more vulnerable to losing these assets and rights due to gender roles and discrimination, separation, divorce, or widowhood, and have less access to capital, extension, inputs, resources, and decision-making processes [21,26]. Women are discriminated again and excluded from resources and decision-making processes due to socio-cultural practices [27–29]. Women's participation in the decision-making processes of various groups has given them a greater



say in accessing common property resources in a society where men have traditionally been the decision makers [30]. Social exclusion and marginalization remain major barriers to micro-level vulnerability planning in Nepal and to the overall development of the poor and marginalized [29]. Local governments and federal and provincial authorities should include activities to promote social inclusion and reduce marginalization in their plans and allocate adequate resources to address key vulnerabilities and climate-related hazards. Local governments, community-based organizations (CBOs), local communities, and households should work together to coordinate, collaborate, and engage in partnerships [8,20]. In addition, socio-economic indicators need to be developed to focus attention on distributive justice issues [31,32], and they need to be realistically measured and regularly reviewed [33]. Inequalities due to gender differences results in differential access to locally available adaptation resources and must take into account adaptive capacities and adaptation barriers [34]. In many situations, this unequal access to adaptive resources can further entrench local inequalities as well as pre-existing vulnerabilities [35]. Gender is a deeply rooted contextual condition that influences vulnerability through interaction with other conditions and socio-economic factors [16].

In 2010, the Government of Nepal adopted the National Adaptation Program of Action (NAPA) and Climate Change Policy. In 2011, Nepal recognized gender as a cross-cutting issue for adaptation plans [23], and the Ministry of Forests and Environment also prepared a gender-responsive Local Adaptation Plan of Action in 2019. In addition, to facilitate NAPA in Nepal, Local Adaptation Plans of Action (LAPAs) are being developed for villages, and CAPAs are being implemented at the community level following the National Framework on Local Adaptation Plans for Action [36]. Both LAPAs and CAPAs are designed through a bottom-up approach and developed in line with the Local Self Governance Act (LSGA), which considers strong provisions for poor, *Dalit*, ethnic, and gender groups [37]. CAPA groups integrate community- and ecosystem-based adaptation planning to minimize climate impacts on livelihoods and to prepare adaptation measures for building resilience capacities of vulnerable people. The impact of climate change on the wellbeing of individuals, households, and groups, and their ability to respond to these changes, depends on the context in which climate change occurs [6]. This context includes all factors, i.e., biophysical characteristics, information and technology, and institutional arrangements, that determine the vulnerability of an individual, household, group, or community to climate change [11,38,39]. The increasing impact of climate variability may challenge men and women differently due to social, economic, cultural, and power relations, as well as access to resources [11,38,40]. The associated disasters and impacts often exacerbate existing inequalities, vulnerabilities, economic poverty, and unequal power relations [11,41]. Adaptation to climate change assumes that economically poor people and socially excluded groups are not only disproportionately affected by vulnerability but also neglected in specific adaptation efforts. Therefore, this work focuses on identifying the status of differential impacts of climate change vulnerability in the community and designing an adaptation plan to respond to the nature and intensity of vulnerability and impacts in different geographical locations of Nepal. The study compares the differential impacts of climate change by wellbeing, caste/ethnicity, and gender in relation to specific climate-related hazards and the exposure, sensitivity, and adaptive capacity of communities. The study examines the most vulnerable groups with differential impacts of climate change on selected livelihoods.

## 2. Materials and Methods

### 2.1. Case Study Site

The study was carried out in three districts: Kailali, Kaski, and Gorkha, representing Tarai, Middle Mountain, and High Mountain regions, respectively (Figure 1). These groups were selected based on the fact that they are the most vulnerable regions, districts, and communities where climate-related hazards have had a significant impact on people's lives and livelihoods over the past 30 years. The study areas were selected with a number of

CAPAs representing high populations of women, *Dalit*, marginalized *Janajati*, and socially excluded groups. About 10% of CAPAs (N = 9) were sampled out of 94 CAPA groups, and about 13% of 566 vulnerable households were sampled. Within each CAPA group, which is a part of the Community Forest User Groups (CFUGs), vulnerable and non-vulnerable groups were stratified into different wealth groups, and then sample households were randomly selected from the vulnerable group. The household was selected as the main unit of study because the most important decisions about climate change adaptation and livelihoods are made at the household level [42]. The CFUG is one of the largest community-based organizations and comprises more than 40% of the total population of Nepal. It comprises more than 17,500 groups and is responsible for managing 1.2 million ha (36% of the total) of forest resources in Nepal [43].

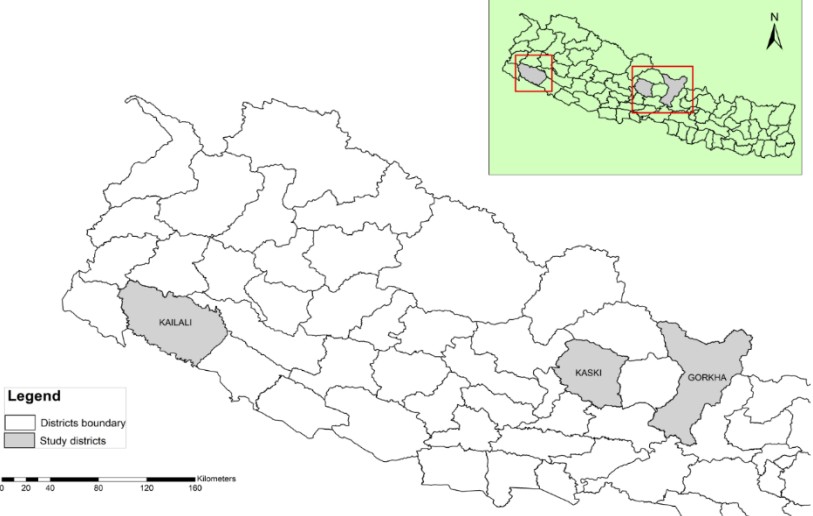

**Figure 1.** Case study area.

*2.2. Research Methods*

We followed a case study approach as it allowed in-depth exploration of knowledge challenges from the perspective of participants [44] and drew on multiple sources of evidence [45]. Using community forest user groups in community forestry as an example, we examined CAPA preparation and implementation processes in three randomly selected districts in Nepal. Climate change is a complex problem that interacts with multiple processes, and using a mixed-methods approach allows for holistic understanding of the different dimensions of the problem [6]. In general, the hypothesis of this study is that the stressors associated with climate change affect poor people, women, men, and marginalized *Janajati* and *Dalit* differently. Women are affected more than men because gender roles require them to spend more time and effort obtaining natural resources such as water, agricultural products, etc.; poor families more than well-off families because extreme weather conditions, water scarcity, and lower productivity threaten their livelihoods, leading to food insecurity; and *Dalits* and *Janajati* more than *Brahmins* and *Chhetri* from the caste/ethnic groups because of traditional occupational activities and limited options for livelihood diversification. Two research questions were developed: (i) To what extent do climate change and variability impact exposure, sensitivity, and adaptive capacity based on gender, wealth, and caste/ethnicity? (ii) What are the challenges for men and women, caste/ethnicity, and economic group to coping and adapting to current climate change and variability? The desk review was followed by direct field observations, 73 household interviews, 13 key informant interviews, consultations, and 9 interactive meetings in 3 districts. Where affluent groups were under-represented, key informants were used to identify suitable households to supplement the sample. In the study districts, a participatory wealth ranking was derived to categorize the study households into three wealth strata (well-off, middle, or poor) based on the relative wealth position of the households in the community

using local criteria of wealth [46]. A stratified random sampling procedure representing all wealth groups was used to select respondents for the in-depth interview. At least one interactive meeting was conducted with each CAPA group with 12 and 30 stakeholders of different socio-cultural representations to further explore natural hazards and disaster impact on livelihood activities.

### 2.3. Differential Impacts of Climate Change

For the study, the International Food Policy Research Institute (IFPRI) framework was used to examine the different impacts and responses by different actors to climate change [39]. The framework was placed in a vulnerability context that included a number of interrelated factors/elements, such as user characteristics, biophysical characteristics, information and technology, and institutional arrangements. For the purpose of the study, the framework was slightly modified in terms of different actors, i.e., from two actors to five actors: women, men, *Dalit*, marginalized *Janajati*, and wellbeing in adaptive capacity and resilient communities/groups and households. These five actors or groups were analyzed to explore the vulnerability contexts that may differentially influence the decision-making process in terms of exposure, sensitivity, and adaptive capacity. The study also focused on the decision-making power of different actors over access to and control over available resources and services that relate to adaptive capacity and contribute to resilience building of individuals, households, and groups. In this paper, multivariate data on household exposure, sensitivity, and adaptive capacity were analyzed using canonical correspondence analysis (CCA) in Canoco 5 [47]. The questions were divided into eight groups, which were later analyzed separately (Table 1). The main elements of vulnerability contexts, User characteristics, Information and technology, Institutional capacity, and Biophysical, were matched with those mentioned in the literature [6,11,39,48]. The table shows the main questions related to vulnerability contexts with tracking of exposure, sensitivity, and adaptive capacity in terms of household ability (further tested by gender, wealth, and caste/ethnicity) to adapt to climate change and variability in order to use this information to develop CAPAs at the household and community levels. The categories of gender, caste/ethnicity, and wellbeing were used as explanatory variables in all analyses, and the significance of their effects was tested by the Monte Carlo permutation test. For the contexts at risk, the location of the settlement in relation to risk (on a scale of low to high) was analyzed separately as a function of wellbeing, caste, and gender using multiple regressions with R 3.2.0 [49].

**Table 1.** Indicators of household livelihood vulnerability collected through a household survey in nine CAPAs groups in Nepal.

| Vulnerability Contexts | Vulnerability Factors | Indicators | Sub-Indicators | Questions Posed During Household Survey |
|---|---|---|---|---|
| User characteristics | Exposure | Geography | Degree of risk of settlements | 1. What is the degree of existing location of settlements/households? High, Medium and Low? |
| | | Natural disasters | Lists and average numbers of climate-induced hazard for the last 30 years | 2. What have been the climatic changes in your locality in the last 30 years? |
| Information and technology | Sensitivity | Ownership of communication media | Number of available communication devices: television, radio, and mobile | 3. Do you have access on the weather forecast information?; What communication media do you have? e.g. television, radio, mobile phone; Do you have knowledge of early warning system, Vulnerability mapping, alternative options and activities to these activities to manage CC impacts? Yes, No or Don't know |
| | | Access and Knowledge | Weather forecast information, early warning system, vulnerability mapping, alternative options, and activities for coping with CC impacts | |
| Institutional arrangement | Adaptive capacity | Decision-making process | Committee and general assembly activities: deciding date and venue, setting agenda, information sharing with users, and making decisions | 4. How are committee meetings and assembly of community forest user groups called and decided, and who decides or has responsibility? Committee, chairman, secretary, communities and forest guards or if any. |
| | | Bargain power and access to market and other resources | Access to market and market facilities, local norms/rules to use the resources | 5. Is there a local market? Do you have access to the market? Are there local norms/rules regarding the use of resources? Yes, No and who decides on the use of resources? Committee, chairperson, secretary or community or if any. |
| Biophysical characteristics | Sensitivity/Adaptive capacity | Livelihood options | Family members responsible for firewood collection, domestic work, cultivation of land, and selling of land, agricultural products, livestock, vegetable products, and fruits | 6. Who is responsible for firewood collection, fetching the water, domestic work; cultivate land, selling land, agricultural products, livestock, vegetable products and fruits? Men, women or both |
| | | Livelihood strategies and their prioritization | Degree of workload, dependence on selling food crops, access to income, control of income, and prioritization for planting and crop diversification | 7. What is the workload situation after community forest management and fetching natural resources? Who has more ownership for food marketing, access to income? And control over income? Who has more priority for planting and crop diversification? Women, men or both |
| | | Social network and inclusiveness | Provision of social inclusion in CAPA documents for poor, women, and excluded groups | 8. Does the CAPAs document plan address the needs of the poor, women and excluded groups? Yes, No, Don't know |

## 3. Results

### 3.1. User Characteristics

In this study, the respondents' responses were mainly categorized to examine the vulnerability context and user characteristics of the respondent households tested by gender, caste/ethnicity, and economic groups in terms of exposure, sensitivity, and adaptive capacity. Exposure was measured in terms of the impact of climate-related vulnerability, such as the location of the settlement in relation to the level of risk and experience of climate-related hazards.

#### 3.1.1. Location of Settlement in Terms of Risk

From the documents of CAPA, available participatory and resource maps, and direct observation by the research participants, the most vulnerable households in the community were confirmed. The results showed that vulnerability factors such as exposure varied according to the wealth status of the respondents. The location of the settlement in relation to risk is the critical factor in distinguishing wealthy from poor and very poor households of the respondents (parameter estimate = $-0.1968$, t = $-2.277$, $p = 0.0258$), but the difference was not very significant. In the hilly regions, e.g., Kaski and Gorkha districts, most of the very poor households lived in steep and landslide prone areas. There was no significant difference or consistent pattern between the categories of caste/ethnicity and location of the household; rather, it was random. It is possible that people perceive their risk targets and values (e.g., exposure) and potential livelihood benefits, services, and opportunities (e.g., sensitivity and adaptive capacity) differently. Differences between castes/ethnicities in exposure were not significant and may not be important factors in distinguishing the degree of risk.

#### 3.1.2. Experience of Climate-Related Hazards

Households ($n = 73$) were asked to describe their experiences of climate-related hazards that directly and indirectly affect their normal lives, crop production, livelihoods, and health. Respondents identified a total of twelve different hazards in the sampled districts. Floods, hailstorms, invasive species, cold/heat waves, and drought are the most important climate-related hazards. Respondents' experience of hazards by wealth categories showed that the lowest percentage of wealthy respondents experienced the hazards, but a higher percentage of very poor, poor, and middle-class respondents expressed experiences of flood, hailstorm, landslide, soil erosion, drought, and invasive species. Canonical correspondence analysis (CCA) found that the hazards of storms and severe weather did not differ significantly according to the wellbeing of the respondents. Similarly, respondents from different ethnic and gender groups experienced and witnessed hazards differently (Figure 2). Dalit respondents reported storms as the most common hazards, while floods, cold waves, insects, and diseases were cited as the most common climate-related hazards faced by *Janajati* respondents. Soil erosion and landslides were cited as the main impacts of climate change by affluent and middle-class respondents. Insects and pests, extreme cold, and heat waves were mentioned as major hazards by female respondents, while hailstorms and landslides were mentioned by very poor and poor respondents. *Brahmin/Chhetri* respondents were most likely to experience soil erosion, hailstorms, and floods, while they were least likely to experience storms. Such differential impacts of hazards are directly related to livelihood resources, agricultural productivity, livestock, and communities.

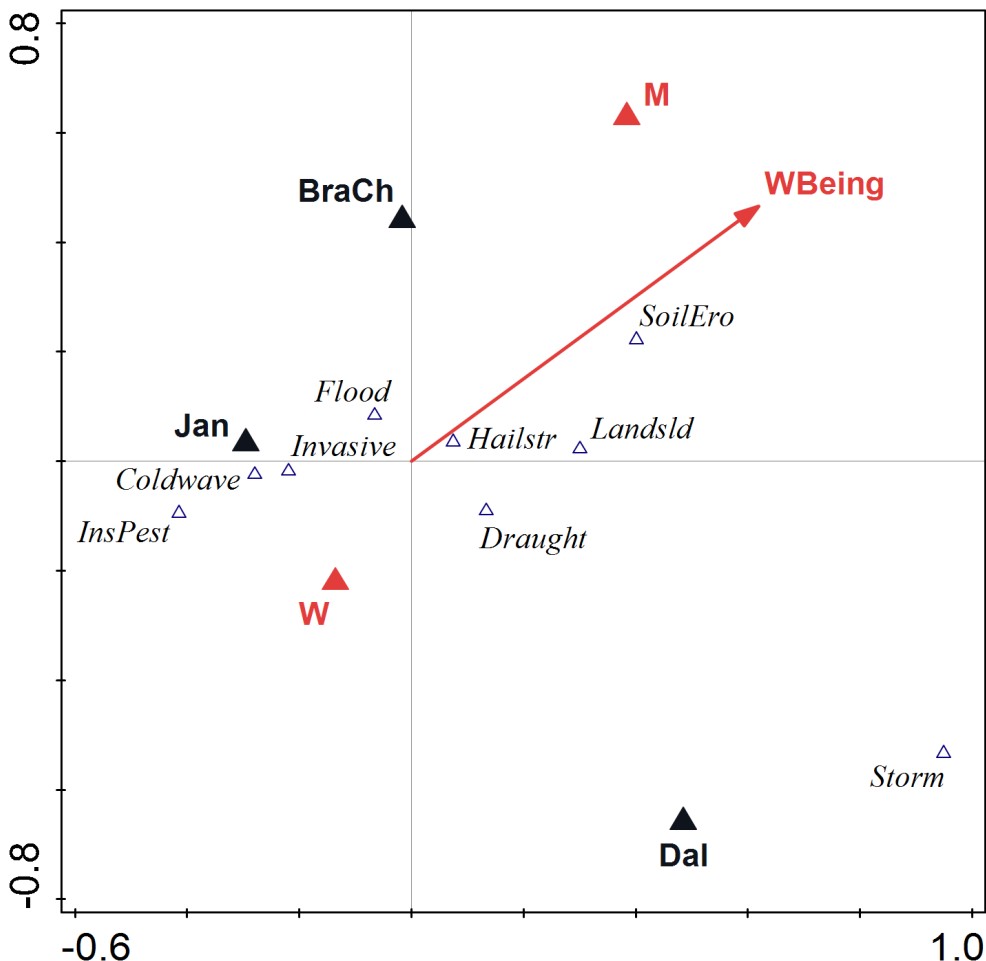

**Figure 2.** Canonical correspondence analysis (CCA) for climate-related hazard experiences based on gender, caste-ethnicity, and wellbeing: gender, men (M) and women (W); caste/ethnicity, Dalit (Dal), Janajati (Jan), and Brahmin/Chhetri (BraCh); flooding (Flood); drought (Draught); landslide (Landsld); cold/heat wave (Coldwave); invasive species (Invasive); insects and pest (Inspest); soil erosion (SoilEro); storm (Storm); and hailstorm (Hailstr).

### 3.2. Information and Technology

Information and technology is another component of the vulnerability context as described in the framework. It is the capacity and type of adaptation response that depends on the access of individuals, households, or groups to information about climate risks and appropriate responses.

Access to and Ownership of Communication Media: Radio, Television, and Mobile Phones

Figure 3 shows that large differences in access to information and communication media by wealth status, caste/ethnicity, and gender are mainly due to: (i) a high percentage of very poor and poor households without information and communication tools such as a TV and mobile phone; (ii) availability of communication media among all affluent families, including male and female family members with mobile access among middle and affluent category respondents; (iii) less or no access to TV, radio or mobile phone among Dalit respondents; (iv) interestingly, few Dalit women have access to mobile phones, as their male partners mostly work outside the villages or abroad; (v) higher percentage of access to radio and television among Brahmin/Chhetri respondents; and (vi) lower percentage or no uniformity of access to information and communication among Janajati as compared to Brahmin/Chhetri (Figure 3).

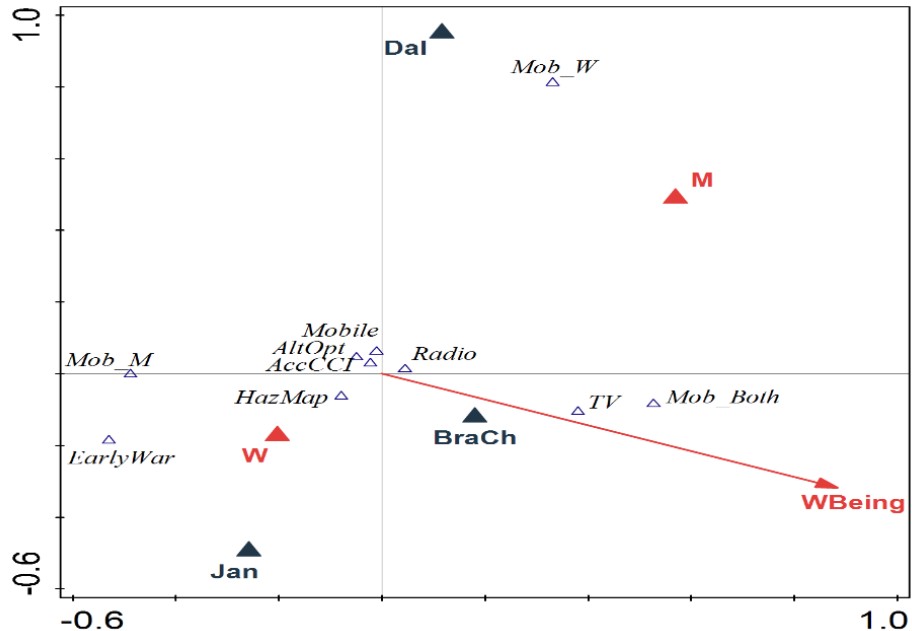

**Figure 3.** Canonical correspondence analysis (CCA for Information and Technology by gender, caste/ethnicity, and wellbeing categories. Communication mode: television (TV); radio (Radio); mobile (Mobile); and knowledge and access from early warning systems (EarlyWar); hazard mapping (HazMap); alternative options (AltOpt); and access to activities for coping with CC impacts (AccCCI).

The differences in access to early warning systems, vulnerability maps, alternative options, and activities to address climate-related impacts were mainly due to: (i) higher percentage of very poor and poor households with knowledge and access to early warning systems due to involvement and participation in the CAPA development processes; (ii) differences in access to vulnerability maps and alternative options among very poor and poor households compared to affluent and middle class respondents; (iii) a higher percentage of *Janajati* reporting access to early warning systems and hazard maps as compared to *Brahmin/Chhetri* and *Dalit* households; (iv) no *Dalit* respondents knew about early warning systems; (v) less access to early warning systems among affluent and middle categories compared to poor and very poor respondents; and (vi) higher percentage of female respondents reporting access to and knowledge about early warning systems, vulnerability maps, and alternative options compared to male respondents. Adaptive capacity to cope with climate change impacts also differed significantly by caste/ethnicity, gender, and wealth categories. The differences were due to: (i) a higher percentage of access to activities to cope with climate-related impacts, e.g., (ii) a higher percentage of knowledge and access to these activities among female respondents from *Janajati* compared to other categories; (iii) lower access to activities to address climate change impacts among very poor and poor households; and (iv) overall, a higher percentage of access to activities to address climate change impacts among female respondents compared to men, which was due to women being involved in the CAPA program and having recently participated in several training events (Figure 4).

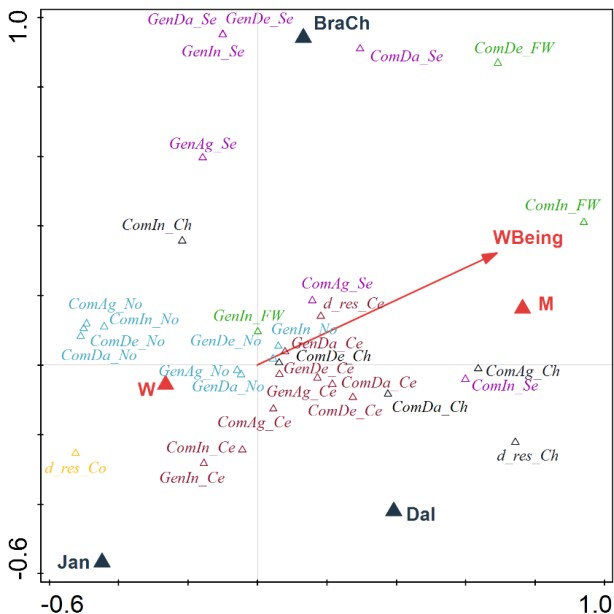

**Figure 4.** Canonical correspondence analysis (CCA) for decision-making process at Committee meeting and General Assembly of forest user groups by gender, caste/ethnicity and wellbeing categories. Chairperson, Ch; Secretary, Se; Committee, Ce; Community, Co; Forest Watcher, FW; and Unknown, No. Committee/General assembly, respectively: date and venue (Comdate or Gendate); setting the agenda (ComAgda or GenAgda); informing users (ComInf or GenInf); and making decisions (ComDec or GenDec).

### 3.3. Institutional Arrangements

Adaptation depends not only on access to resources, information, and biophysical characteristics, but must also be considered in the context of the institutional environment. In these vulnerability contexts, three questions were asked to the respondents: (i) Who are the decision makers for the collective decision-making process in the process of convening and decision making at the committee level and in the general assembly? Who has the bargaining power over access to resources, access to local market, chemical fertilizer, hybrid seeds, improved cook stove and local resource use rules were considered for the analysis of differential impacts? To answer the first question, respondents were asked about four activities (deciding the date and venue, setting the agenda/presenting proposals, informing users and committee members, and making decisions) for the main leaders in the decision-making process in both the committee and plenary meetings.

### 3.3.1. Processes for Calling and Decision-Making in Committee Meetings and Assembly

The differences in respondents' perceptions of the process for convening committee meetings and assemblies and decision-making were related to setting the date and place, setting the agenda, informing users, and decision-making. Canonical correspondence analysis (CCA) of each activity in the decision-making forum for Committee and Assembly are highlighted in different colors (Figure 4). The significant differences in the decision-making process by wellbeing, gender, and caste/ethnicity were mainly due to (i) significant differences in the perception of female respondents who reported being unaware of the information sharing or decision-making process in both the committee and the general assembly as compared to male respondents; (ii) a higher percentage of women indicated that the decision was made by the Chair, Secretary, and the Committee; (iii) a higher percentage of decisions in the Committee meeting and the General Assembly were made by members of the key portfolio, particularly the Chair and Secretary of the forest user groups; (v) a higher percentage of *Brahmin/Chhetri* believed that the decisions regarding the date and venue of the committee meeting, general meeting, agenda of the general meeting, and information about the decision were taken by the Secretary; (vi) a higher percentage of

Dalit stated that the chairperson of the CFUG was responsible for the implementation of the decision, while *Janajati* believed that the community is/was responsible for the implementation of the decision of the committee meeting and general meeting; and (vii) a higher percentage of *Brahmin/Chhetri* were aware of all types of decisions taken by the Committee and General Assembly compared to *Dalit* and *Janajati*. In addition, participants agreed that less access to information and a passive role in decision-making lead to a lack of awareness and ownership of institutional activities, resulting in low adaptive capacity to deal with the impacts of climate change. Differences in adaptive capacity were also due to poor access to the decision-making process by the very poor and poor compared to the affluent and middle class and women in terms of deciding the date and venue, setting the agenda, informing users, and decision-making. Such differences significantly trigger climate change and variability impacts.

### 3.3.2. Perception of Bargaining Power and Access to Markets and Other Resources

The impacts of climate change, variability, and extremes on livelihoods and adaptive capacity is related to bargaining power and access to resources at the individual, household, or community level. The holder of decision-making power over the use of resources, market access, and bargaining power in the market with respect to the sale of products and access to chemical fertilizers, hybrid seeds, and improved cookstoves are important triggering factors for climate change vulnerability. Differences in bargaining power and access to markets and other resources were also due to poor institutional arrangements of the poor and very poor compared to the affluent and middle class and between gender and caste/ethnic groups. The main differences were due to: (i) a higher percentage of women, poor, and very poor categories who reported that their bargaining power increased with access to the market; (ii) a lower percentage of men and women accessing the local market and not being able to get more benefits either through selling or buying products; (iii) a higher percentage of men accessing the local market; (iv) a higher percentage of *Janajati* reporting access to the market and bargaining power compared to *Dalit* and *Brahmin/Chhetri*; (v) no access to the market or bargaining power in the market was reported by *Dalit* respondents; (vi) a higher percentage of middle and affluent categories reported access to improved cookstoves; (vii) no access to hybrid seeds, improved cookstoves, and chemical fertilizer was reported by *Dalit* respondents; and (vii) a higher percentage of *Dalit* respondents considered local norms/rules related to resource use (Figure 5).

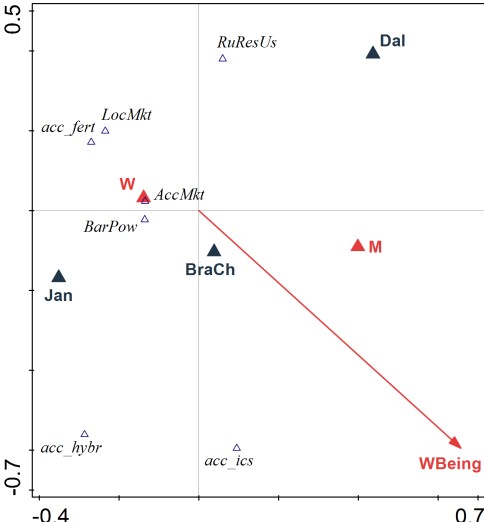

**Figure 5.** Canonical correspondence analysis (CCA) for perception of bargaining power and access to markets and other resources by gender, caste/ethnicity, and wellbeing categories. Market access (AccMar); bargaining power (BarPow); access to local market (LocMkt); access to hybrid seeds (acc_hybr); access to improve cooking stoves (acc_ics); access to chemical fertilizers (acc_fert); and rules of resource use (RuResUs).

*3.4. Biophysical Characteristics*

3.4.1. Differentiation by Responding Livelihood Options

Biophysical characteristics refer to the sensitivity and adaptive capacity of the physical and ecological systems that define the natural limits of climate adaptation. Eight livelihood options involving either men, women, or both were identified and questioned about their responsibilities at the household level. Differences emerged mainly due to: (i) a higher percentage of men from *Brahmin* and *Chhetri* households reported being responsible for cultivating land compared to *Janajati* and *Dalit* households; (ii) a higher percentage of both men and women from *Brahmin* and *Chhetri* households reported being responsible for collecting firewood, household chores, and selling land, agricultural produce, livestock, vegetable products, and fruits; (iii) a higher percentage of *Janajati* women were responsible for collecting firewood and selling livestock compared to *Brahmin/Chhetri* and *Dalit* groups; (iv) a higher percentage of women reported being responsible for selling land, agricultural products, and fruits; (v) there were no consistent patterns among *Dalits* in livelihood options; (vi) a higher percentage of poor and very poor households were responsible for collecting firewood and selling agricultural produce and crops; and (vii) poor and very poor *Brahmin/Chhetri* families reported being responsible for domestic chores and selling livestock, agriculture products, and fruits (Figure 6). These livestock options have a direct impact on adaptation to climate impacts, with a significant difference between gender, wealth, and caste/ethnic groups responsible for selling physical goods. This implies that the impact was greater for poor families and women who are in the frontline and suffer more from the reduction in livelihood options and unequal access to resources and services. The result confirms that the poor, women, and *Dalit/Janajati* within the communities are affected differently by the impacts of climate change and may not be able to easily overcome the impacts of climate change and variability.

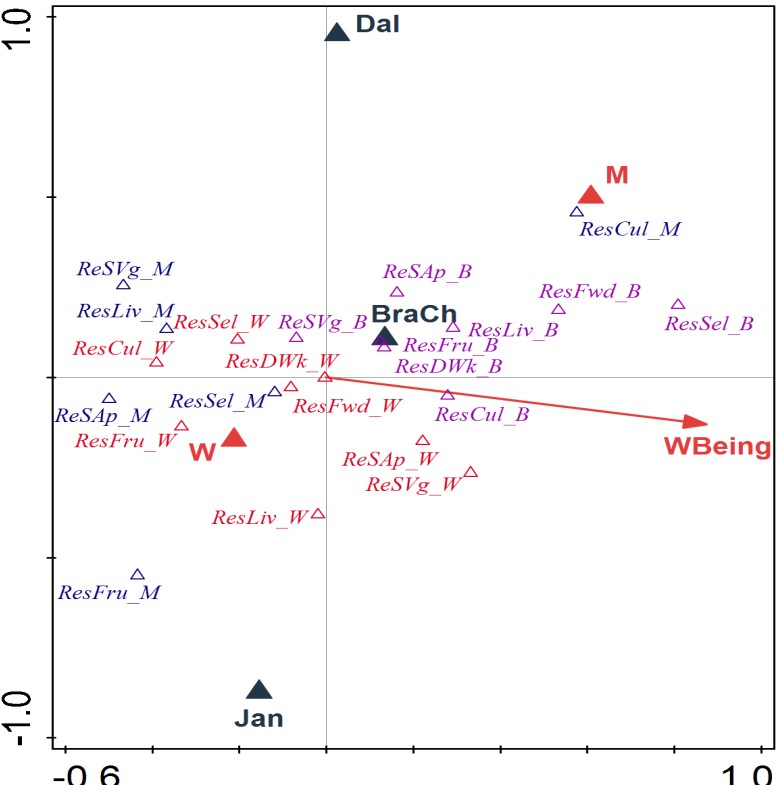

**Figure 6.** Canonical correspondence analysis (CCA) for livelihood response options by gender, caste/ethnicity, and wellbeing categories. Responsible for fuelwood collection (ResFwd); domestic works (ResDWrk); cultivation of land (ResCult); selling land (ResSel); agricultural products (ResSellAp); livestock (ResLivst); vegetable products (ResSelVg); and fruits (ResFrut).

### 3.4.2. Differentiation by Livelihood Strategies and Prioritization

Differences in sensitivity and adaptive capacity in terms of food crops, income, plantation and cropping diversification, and workload situations are important factors in defining the degree of vulnerability. In the context of climate change and its impacts, access to and control over income may be particularly important for the poor and women, as these help the household cope with hardship in order to afford needed materials. Following the framework of the study, respondents were asked a series of questions about the decision-making power of different actors over access to and control over income, markets, resources, and services. The interrelationships between and among the contexts of vulnerability and adaptation were analyzed to enhance adaptive capacity and contribute to resilience building. Differences were due to: (i) higher percentage of access and control over income reported by men, affluent families, and *Brahmin/Chhetri* groups; (ii) higher percentage of control over income and food marketing reported by *Janajati*, while women's workload decreased in *Janajati* groups; (iii) lower percentage of access to income, control over income, and planting and crop diversification activities among Dalit respondents, while both *Dalit* men and women have access to food marketing; (iv) steady workload or no change in workload of women in affluent groups, while workload increased in poor and very poor households; (v) reduced workload of men and *Brahmin/Chhetri* groups; (vi) female Janajati reported that men control income, marketing of food crops, planting of trees, and diversification of crops as compared to Dalit and Brahmin groups; and (vii) increased workload of women in *Dalit* and *Janajati* groups as compared to *Brahmin/Chhetri* caste/ethnic groups (Figure 7).

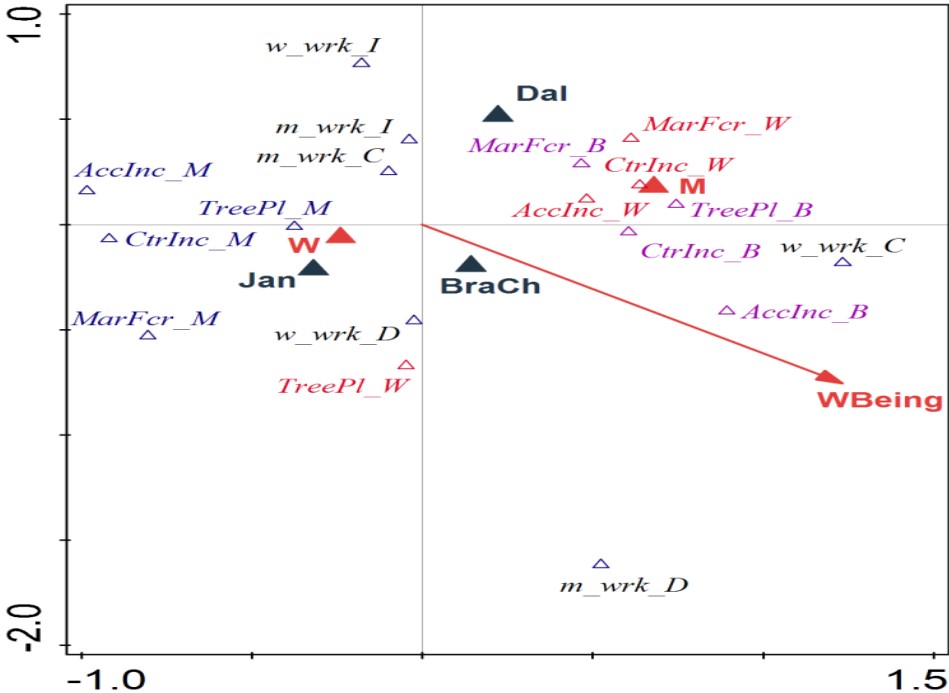

**Figure 7.** Canonical correspondence analysis (CCA) for adaptive capacity by gender, caste/ethnicity, and wellbeing categories. Access to income (AccInc); control over income (CtrInc); planting and diversifying crops (TreePl); marketing of food crops (MarFcr); and workload with three options (decreased, increased, or constant) for both men and women (m_wrk_D or I or C; w_ wrk_D or I or C; or B_wrk_D or I or C, respectively).

### 3.4.3. Adaptation Arena: Social Networks and Inclusiveness

The last section of the differentiated analysis raises the question of whether special provisions for women, excluded, and poor populations are guaranteed in the context of adaptation decisions in CAPA documents. It is an important adaptation strategy if there are special provisions for weaker sections of society that enhance resilience building

against risks of climate variability and change, such as livelihood diversification. This analysis sheds light on the level of knowledge of inclusion approaches of CAPA groups to address climate change vulnerability, resilience building, and adaptive capacity. Differences emerged from: (i) a higher percentage of *Dalit* respondents were unaware of special schemes from CAPA; (ii) a higher percentage of female respondents were unaware of special schemes except for the socially excluded groups; (iii) a higher percentage of male *Janajati* were unaware of special provisions for women, poor, and socially excluded groups; and (iv) there was a lower percentage of knowledge about special schemes among poor and very poor groups as compared to middle and well-off categories (Figure 8).

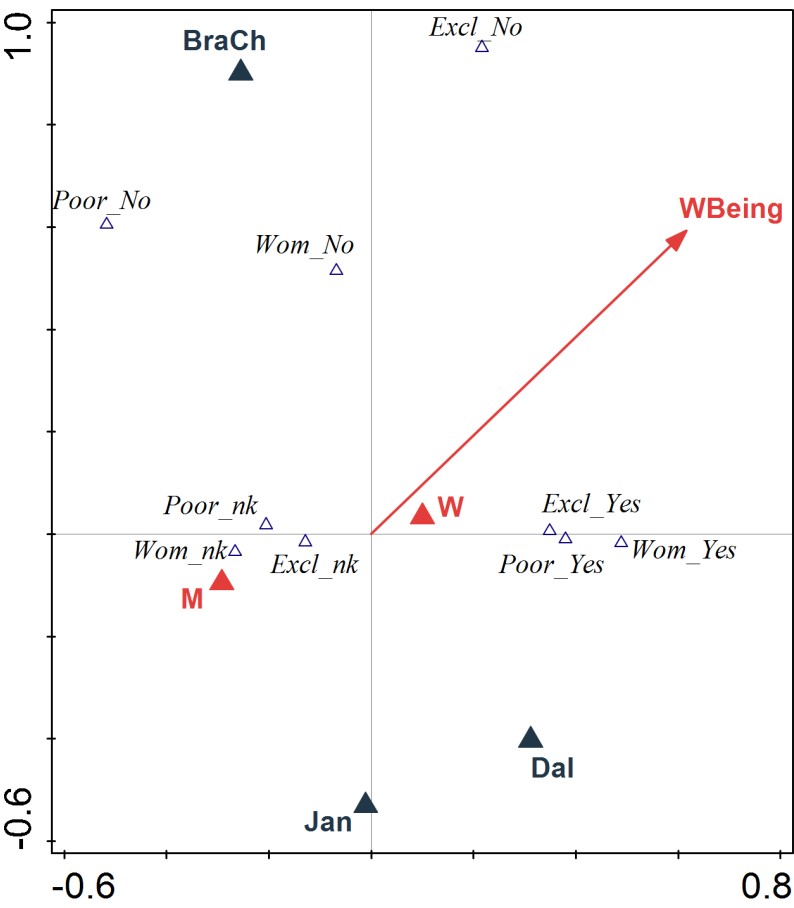

**Figure 8.** Canonical correspondence analysis (CCA) for knowledge of special provisions at CAPAs by gender, caste/ethnicity, and wellbeing categories. Special provision of socially excluded groups: Yes or No or Unknown, respectively, (*Excl_Yes or No or nk*) for women (*Wom_Yes or No*) and for Poor (*Poor_Yes or No or nk*).

## 4. Discussion

Communities are not uniformly vulnerable to climate change, variability, and extremes not only because of social and economic matters, but also due to different exposures to climate-related hazards. It is essential to understand why communities and peoples are disproportionately exposed to and affected by climate-related threats. Differential impact assessment does not generally lend itself to the development of adaptation policies, in part because of uncertainty about the future and socio-economic conditions as well as the broader developmental context [50]. The relationship between vulnerability reduction and poverty reduction is nonlinear and poses challenges for both national climate policy and development policy [15]. Local-scale climate-related vulnerability to climate change and adaptive capacity of local communities respond to climate-related impacts. Those who have resources in only one place are susceptible when it comes to managing vulnerabil-

ities/risks, but marginalized and poor populations that do not have access to resources usually have difficulty managing risks due to social, economic, and political structures. In this context, hazards such as floods and droughts are more severe disasters due to their lack of adaptive capacity to protect themselves from various risks. Unequal distribution of available resources and adaptive capacity is related to adaptation to climate-related impacts that directly affect the control of their daily lives. Analysis shows that women and men, caste/ethnic groups, and wellbeing groups are differently impacted due to climate variability and livelihood options. Women and men perceive experiences of climate change differently in places affected by climate-related hazards such as floods, drought, soil erosion, landslides, and cold waves. Perceptions of the experience of climate-related impacts show that many women are preoccupied with the immediate livelihood options of their households. Therefore, not all women and not all men react to climate change in the same way [25]. In order to understand the role of women in climate change adaptation, it is particularly necessary to understand the power relations between and among women and men, and the ways in which climate change may exacerbate and expand these relations [41]. Economically poor and marginalized households are affected because they often have access only to marginal land, such as land near rivers that is more vulnerable to flooding [51]. Effective local adaptation planning and local engagement for effective adaptation planning by assessing local context, resources, information, communication mode, and institutional capacity will meaningfully contribute to adaptive responses to climate change and hazards.

Vulnerability assessment requires a good understanding of institutions and roles in resource distribution and enforcement of rights and regulations for management [13]. Requiring a climate change system, impacts, vulnerability, and readiness to assess the status, changes, trends of key resources, and understanding climate impacts are urgently needed to support decisions and implementation of the Adaptation Plan Strategy [7,52]. Therefore, to reduce the current differential vulnerability among CAPA groups in Nepal, which is fundamental to building resilience and enhancing adaptive capacity, institutions can provide economically poor, female, and marginalized groups with equal access to livelihood and livelihood diversification options, including information, knowledge, and other resources [7]. Women play an important role in natural resource and environmental management, but their contribution is ignored. From differential exposure analysis, it was found that poor economic groups and *Dalit* and *Janajati* groups mostly live in high-risk conditions, resulting in risks to lives and livelihoods and requiring adaptation measures to address climate impacts [20]. However, women and poor communities usually have limited access to resources, markets, and participation in decision-making processes in the main portfolio of community forest management. There has been limited scope to enhance their adaptive capacity and resilience to climate change risks. Bargaining power to diversify livelihood options, access to information, technology, and decision making are key processes to increase access to and control over income. These provide economic opportunities for poor women and ethnic and marginalized groups. The impacts of hazards reported by communities varied according to the wellbeing status, geographical location, and occupation of the communities, but the links between the impacts of climate change and the socio-political structure of society need to be analyzed in depth [5].

However, the vulnerability situation does have gender-differentiated implications due to different gender norms, relations, and cultures and, even though women have a lower adaptive capacity than men, both men and women play an equal role in implementing adaptation strategies; this highly impacts the risk capacities of vulnerable and marginal communities to enhance their self-confidence and adaptive power [16]. A community-based adaptation planning process in Nepal promotes women's representation and leadership. This may also be possible through husband migration, where women have the opportunity to increase their representation and participation. This provides opportunities and space to gain new knowledge, experience, skills, and responsibilities in newly defined areas, as well as more opportunities for social networking and greater bargaining power [53]. The role and responsibilities of women in nuclear families are different and motivate them

to change their position and take more responsibility for additional tasks when men are not at home and possibly influence decision making within the existing socio-cultural structure. It provides them with skills and opportunities in new areas and ultimately helps to build their confidence [20,54,55]. Understanding the barriers to access women face is especially critical, as they are already in an extremely vulnerable situation, and these differences in vulnerability and access to resources that are critical for climate change adaptation intersect with existing social, cultural, and economic determinants [34]. This has been confirmed by [56–58], which explain that the non-adoption of certain agricultural technologies by women was a result of their limited access to information, land, and other important resources due to socio-cultural norms. Women had less access to social amenities, which made them more vulnerable to manifested climate change impacts such as floods and drought compared to their male counterparts [59].

Climate adaptation can be complex, and there is no one-size-fits-all solution [60]. Although climate vulnerability varies by community location, occupation, and gender of the household head, overall vulnerability varies according to the wellbeing status within communities [5]. Low-income households are still the most vulnerable without mobile phone facilities and opportunities because they do not have access to information in case of natural disasters; however, they could get information from television and radio. The female respondents reported that they often work in agriculture or are away from home for various tasks and are not able to watch television or listen to radio programs regularly. The above study discussion shows that some *Dalit* and occupational groups are excluded from all kinds of communication modes as compared to the poor *Brahmin/Chhetri*. Women are busier with domestic responsibilities, some of which limit their time to listen to the radio or watch TV, and they receive secondhand information from neighbors, friends, and family because more men have ready access to information about climate change than women, except for information they receive from friends, family, and neighbors [8,20]. The level of accessibility, including transportation, communication, and access to resources is linked to other functions of production and services and creates a decision space to reduce the impact of climate and adaptation strategies. Gender inequalities, including cultural, economic, social, political, health, and environmental factors, are a key concern in adaptation efforts relevant to resilience and adaptive capacity [61,62]. Critical factors that can help with gender-sensitive adaptation include improved access and ownership of land, microcredit, and enterprises that target women and poor segments of society. Adaptation measures can help through light infrastructure to improve water use and drinking water supply, livelihood diversification through irrigation schemes, provision of quality seeds along with agricultural and crop inputs, storage, education, markets, and climate-smart technology. All of this must be culturally appropriate, socially acceptable, responsive, and practical for women and other vulnerable groups [51,61,62]. Standardized metrics of adaptive capacity are important for designing effective strategies to promote resilience in natural resource-dependent communities and for understanding how social and ecological aspects of communities interact to influence responses [63]. To reduce vulnerability and increase adaptive capacity and resilience of communities, especially women, poor, and ethnic groups of Nepal, sustainable integrated adaptation strategies should be incorporated in the implementation of CAPA in light of existing inequalities of climate change and variability. The authors of [64–66], emphasized that raising awareness, conservation of natural resources, diversification, building physical infrastructure, strengthening grassroots institutions, and mainstreaming climate adaptation into development policies are critical for climate-resilient pathways.

## 5. Conclusions

Based on the perceptions of different stakeholder groups, interactive discussions, and key informant interviews, the results are mixed and contextual. However, there is limited in-depth analysis of these dimensions and the implications of climate change in the development of CAPA documents. It has been challenging to draw conclusions and

predictions and to effectively compare and contrast the case studies and their impacts at the local level. Climate-related impacts must also consider the current rapidly changing social and political processes in the country. In implementing CAPA and other interventions, roles and responsibilities are changing rapidly due to access to information, technology, natural resources, power, and social resilience. Specific arrangements for gender, wellbeing, and socially excluded groups are highly affected by different vulnerability contexts. Based on the findings, it can be concluded that women are disproportionately vulnerable to climate change because, on the one hand, they are more dependent on climate-sensitive resources and, on the other hand, they are more likely to be found in multiple poor situations and are more negatively affected in terms of their assets and wellbeing. *Dalit* and marginalized *Janajati* groups have less access to and ownership of communication media and access to and knowledge of markets and resources, and therefore have more limited capacity, information, and knowledge to engage in the climate change adaptation process. Discrimination in social processes, exposure, and access to and control over natural resources and biophysical assets is quite high within caste/ethnic groups for various reasons, including power distribution in decision making.

The study highlights the asymmetric relationships in vulnerability contexts between and among social and economic groups. Demographics, diversification of livelihood options, socio-economic and cultural values, infrastructure, and technology are changing rapidly, and the role of gender and its impact on vulnerable communities varies by context and location. Case studies show that women and socially excluded groups typically have disproportionately low access to resources, information, and technology—specifically, limited access to weather forecast information and markets and knowledge of early warning systems. Other components of sensitivity and factors of adaptive capacity are severely limited. It is therefore evident that women, poor communities, marginalized *Janajati* groups, and *Dalits* are likely to be most affected by such extreme climate events. The impacts of climate change are not evenly distributed, with the poorest people suffering the earliest and most severe adverse impacts. Aspects of exposure that are related to geography and may be relevant to climate adaptation have not been adequately tapped or assessed, as it is difficult to generalize and compare different aspects of individual case studies. Analysis of the components of vulnerability and adaptive capacity of vulnerability contexts is more useful for designing local adaptation plans. Priority areas for adaptation are particularly required for poorer, ethnic, and resource-dependent communities. Adaptive capacities vary between different ethnic groups, economic capacity, and levels of environment stress, whereas adaptation strategies would benefit from a more systematic differential impact analysis, including improving institutions, local empowerment in decision making, incorporating climate adaptation in local planning, and enhancing technology and communication channels as required with anticipatory actions and projections of climate scenarios.

**Author Contributions:** C.K. wrote the original draft. C.K. and P.C. supervised the project. A.U., N.D. and M.E.-J. performed the methodology of the manuscript. S.B. and C.K. conducted review and editing. A.U. and C.K. coordinated, integrated, and consolidated all resources. M.E.-J. and A.U. performed validation of data and information. N.D. and C.K. performed visualization of data and information. All authors have read and agreed to the published version of the manuscript.

**Funding:** This work was supported by the Ministry of Education, Youth and Sports of CR within the CzeCOS program, grant number LM2018123.

**Institutional Review Board Statement:** The research of our manuscript was privately funded and received no external funding. All data and information were delivered according to the private research of each author. There is no conflict of interest in our manuscript. All subjects gave their informed consent for inclusion before they participated in the study in accordance with the Declaration of Helsinki.

**Informed Consent Statement:** Informed consent was obtained from all subjects involved in the study.

**Data Availability Statement:** The dataset used and/or analyzed during the current study is available from the corresponding authors on reasonable request.

**Acknowledgments:** The authors would like to thank the anonymous reviewers for their valuable comments that helped shape the key messages of the manuscript. Thanks go to the survey participants for sharing valuable information and feedback during the workshops in Gorkha, Chitwan, and Kathmandu. We thank the team at NORMS/Nepal for the field study and the CzechGlobe CZ.1.05/1.1.00/02.0073 project for the assistance.

**Conflicts of Interest:** The authors declare no conflict of interest.

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
