# Peer review of "Differential Impact Analysis for Climate Change Adaptation: A Case Study from Nepal"

_sustainability, doi:10.3390/su14169825_

Round 1
Reviewer 1 Report
Dear authors
I have made many minor comments about the use and misuse of climate change concepts. You should carefully check them and follow them consistently. Besides, you should add some recent pieces of literature about vulnerability; for instance, your reference for vulnerability elements is the IPCC AR4 2007....there are IPCC AR5 2013 and 6 2021 and 2022
Generally speaking, you must improve consistency and coherence in the use of the terms change variability hazards - as indicated in the annotated PDF

Author Response
Dear Reviewer,
Thank you for the opportunity to address comments and suggestions on the manuscript "Differential Impacts Analysis for Climate Change Adaptation: A Case Study from Nepal" (submission # 1820218). We would like to thank you for your comments and suggestions. We feel that we have significantly improved the manuscript with regard to the comments and suggestions. A point-by-point response is provided in attached file.
We hope you and the reviewers will find this manuscript a significant improvement and consider it suitable for publication.

Reviewer 2 Report
No specific comments. I would suggest you to reinforce some sentences by the introduction of the "local authorities" which as you mention in the abstract are key to address the climate change challenges.
Author Response
Dear Reviewer,
Thank you for the opportunity to address comments and suggestions on the manuscript "Differential Impacts Analysis for Climate Change Adaptation: A Case Study from Nepal" (submission # 1820218). We would like to thank you for your comments and suggestions. Thank you so much for your overall suggestion. We have included key local authorities responsible for addressing the differential impact of climate-related hazards.
We hope you and the reviewers will find this manuscript a significant improvement and consider it suitable for publication.

Round 2
Reviewer 1 Report
You have improved the paper following my suggestion.
...women suffer more than men from the impacts of extreme events related to climate change-related induced............
You should delete one word in P 3 lines 99-101 related or induced (not both)
alternatives
climate-change-related or climate change-induced
Author Response
Dear Reviewer,
Thanks for comments. I have checked and corrected as per your suggestions.
